# Atomic-Scale Revelation of Voltage-Modulated Electrochemical Corrosion Mechanism in 4H-SiC Substrate

**DOI:** 10.3390/mi16101129

**Published:** 2025-09-30

**Authors:** Qiufa Luo, Dianlong Lin, Jing Lu, Congming Ke, Zige Tian, Feng Jiang, Jianhui Zhu, Hui Huang

**Affiliations:** 1Institute of Manufacturing Engineering, Huaqiao University, Xiamen 361021, China; 2National & Local Joint Engineering Research Center for Intelligent Manufacturing Technology of Brittle Materials Products, Huaqiao University, Xiamen 361021, China; 3National Key Laboratory of High Performance Tools, Xiamen 361021, China; 4College of Mechanical Engineering, Baoji University of Arts and Sciences, Baoji 721013, China; 5Zhengzhou Abrasives Grinding Research Institute Co., Ltd., Zhengzhou 450007, China; 6Sinomach Diamond (Henan) Co., Ltd., Zhengzhou 450008, China

**Keywords:** SiC substrate, electrochemical corrosion behavior, molecular dynamics simulation, reaction mechanism, modified layer

## Abstract

Electrochemical mechanical polishing is a critical technology for improving the surface quality of silicon carbide (SiC) substrates. However, the fundamental electrochemical corrosion mechanism of the SiC substrate remains incompletely understood. In this study, the electrochemical corrosion behavior of the SiC substrate is explored through comprehensive experiments and molecular dynamics simulations. Key findings demonstrated that the 4H-0° SiC exhibited the highest corrosion rate in a 0.6 mol/L NaCl electrolyte. The corrosion rate increased as the voltage rose within the range of 2 to 20 V. When the voltage was between 20 and 25 V, the system entered the stable passivation region, while when the voltage was 25 to 30 V, partial dissolution of the surface oxide layer occurred. Molecular dynamics simulations further revealed that both amorphization degree and reaction depth on the SiC surface showed a decreasing trend at elevated voltages, suggesting a corresponding reduction in the corrosion rate when the voltage exceeded the optimal range. OH^−^, O^2−^, and •OH generated by the electrolysis of water during electrochemical corrosion would rapidly react with the surface of the SiC anode, and subsequently form a SiO_2_ modified layer. Moreover, these atomistic insights establish a scientific foundation for achieving superior surface integrity in large-diameter SiC substrates through optimized electrochemical mechanical polishing processes.

## 1. Introduction

Silicon carbide (SiC), as a prominent third-generation semiconductor material [1,2,3], has become indispensable in advanced technological applications owing to its excellent physical and chemical properties, including superior thermal conductivity, remarkable chemical stability, wide band gap, and high critical breakdown electric field [4,5,6,7]. In SiC device fabrication, substrate surface quality critically influences the device performance, operational reliability, and service lifetime [8,9,10]. Surface polishing, being a crucial manufacturing step for SiC substrate, aims to obtain an atomically smooth and damage-free surface to satisfy the stringent requirements of high-performance devices [11,12,13]. Nevertheless, the intrinsic hardness (Mohs 9.5) and brittleness of SiC presented significant challenges for conventional polishing methods in simultaneously achieving high material removal rates and superior surface quality [14,15,16].

To address the inherent limitations of conventional polishing methods, researchers have actively pursued innovative approaches for SiC substrate processing [17,18,19,20,21]. The electrochemical mechanical polishing (ECMP) technique has demonstrated particular promise, offering distinct advantages over traditional methods [22,23,24,25]. ECMP introduced a controlled DC electric field to conventional chemical mechanical polishing systems, enabling precise modulation of surface electrochemical reactions. The applied potential facilitated the electrochemical oxidation of the SiC surface, generating a softened oxide layer (typically SiO_2_) that significantly reduced surface hardness. This synergistic effect between electrochemical modification and mechanical abrasion substantially enhanced material removal efficiency while maintaining superior surface quality, achieving surface roughness values below 0.3 nm in optimized conditions [26,27,28]. A fundamental understanding of the electrochemical corrosion mechanism in SiC substrate is crucial for ECMP optimization, as it governs both the material removal behavior and regulation of the electrochemical reaction during the polishing process.

Current research on SiC ECMP technology has made significant progress in optimizing process parameters, including electrolyte compositions [29,30], abrasive characteristics [31,32], and polishing pad materials [33,34]. However, the fundamental electrochemical corrosion mechanism of the SiC substrate remains incompletely understood. Through linear sweep voltammetry and anodic oxidation experiments, Yang et al. [29] demonstrated that etch pit formation on 4H-SiC(0001) surfaces correlates with localized breakdown sites during anodic oxidation. Subsequent work by Yang et al. [35] characterized the anodic oxidation behavior of 4H-SiC(0001) in NaCl aqueous solutions, developing an electrochemical impedance model to explain surface-dependent oxidation variations. Atomic force microscopy observations by Yang et al. [36] revealed that ECMP-induced oxidation initiated preferentially at step edges, with subsequent oxide bump formation leading to increased surface roughness. Further investigations identified that charge utilization efficiency during 4H-SiC(0001) polishing was limited by current density, directly suppressing the material removal rate [37]. Molecular dynamics simulation serves as an effective means for investigating atomic-scale corrosion mechanisms at SiC interfaces. Tian et al. [38] employed molecular dynamics to examine the material removal mechanism during scratching of 4H/6H-SiC, demonstrating that the C-face exhibits higher removal efficiency and reduced amorphization compared to the Si-face, attributed to distinct dislocation behaviors on the basal plane. Xie et al. [39] conducted reactive molecular dynamics simulations of SiC oxidation across 300–2300 K, identifying three characteristic stages: initial low-temperature adsorption, intermediate rapid oxidation, and final high-temperature interfacial reaction. Their study established a two-stage kinetic model for oxide layer growth, revealing temperature-dependent activation energy barriers associated with oxygen diffusion. Shi et al. [40] investigated the water corrosion mechanism of SiC through reactive molecular dynamics simulation, identifying three temperature-dependent regimes: surface groups formation without corrosion (<1000 K), proportional volatilization of C and Si atoms (1000–1500 K), and formation of a Si-O-Si network that inhibited Si loss (>1500 K). While these studies provided valuable insights, the underlying mechanisms governing high-voltage electrochemical corrosion of the SiC substrate still require comprehensive elucidation.

This study systematically investigated the electrochemical corrosion behavior of SiC substrate through integrated experimental characterization and reactive molecular dynamics simulations. A comprehensive analysis was conducted to elucidate the electrochemical corrosion mechanisms by examining three key variables: (1) crystal polytype variations, (2) electrolyte concentration effects, and (3) applied voltage dependencies, complemented by reactive molecular dynamics simulations of interfacial electrochemical processes.

## 2. Experimental

### 2.1. Experimental Details

The experimental setup of the electrochemical workstation is shown in Figure 1. The electrochemical measurements were performed using a Wuhan Corrtest CS310M workstation with a potential control range of ±10 V and a current control range of ±2 A. The potentiodynamic polarization measurements were conducted at a scan rate of 10 mV/s. For static corrosion tests requiring higher potentials, a DC regulated power supply with a potential range of 0–60 V and a current range of 0–5 A was employed. A standard three-electrode configuration was adopted: (1) A SiC sample secured by a platinum sheet electrode clip served as the working electrode, (2) a 10 × 10 × 0.2 mm platinum sheet as the counter electrode, and (3) a silver/silver chloride (Ag/AgCl) electrode as the reference electrode. The SiC samples with a diameter of 50.8 mm were supplied by TankeBlue Co., Ltd., including 4H-0°, 4H-4°, 6H-0°, and 6H-4° types. Experiments were conducted on the Si-face (0001) of SiC samples, with an initial surface roughness of approximately 1.5 nm (Ra) and no obvious oxidation on the surface. All experiments employed NaCl solution as the electrolyte.

The surface morphology of SiC samples after electrochemical corrosion was observed by a three-dimensional video microscope (KH8700, HIROX, Tokyo, Japan). Surface roughness measurement was conducted with a white light interferometer (NewView 7300, ZYGO, Middlefield, OH, USA), employing Ra as the evaluation metric. A scanning area of 140 × 105 µm was analyzed, with Ra values measured at nine uniformly distributed positions on the Si face of the SiC substrate. The final roughness values represented the mean and standard deviation of these measurements. Chemical composition analysis before and after corrosion was performed using a scanning probe micro-Raman spectrometer (Alpha 300RA) with the following parameters: 532 nm laser (100 mW output power, 100% laser power retention), a 1800 grating frequency (500 nm), 1.3 µm spot size, 50× objective magnification, and a Raman shift range of 0–1200 cm^−1^. Point scans were acquired at the processed regions of the SiC substrate, with peak calibration referenced to the 520 cm^−1^ Raman band of single-crystal silicon.

### 2.2. Reactive Molecular Dynamics Simulation Details

To deeply explore the nanoscale electrochemical interaction mechanism at the SiC interface, a reactive molecular dynamics simulation was employed to resolve atomic-level reaction details and interfacial evolution. In this study, the Lammps software (Version number: lammps-64bit-11Aug2017-MPI) was used to construct a SiC crystal structure model. According to the lattice parameters of the 4H-SiC unit cell (a = 3.076 Å, c = 10.053 Å), a 4H-SiC unit cell structure with an inclination angle of 0° and a Si face on the upper surface was precisely constructed, and a SiC model was formed through periodic expansion. Water molecules were added to the upper surface of the SiC model to simulate the interfacial electrochemical reaction environment, and a fixed layer and a constant temperature layer were set on the lower surface of the model. The simulation model is shown in Figure 2.

For the atomic types in the simulation system, the ReaxFF reactive force field was selected to describe the interactions between atoms. The NVE ensemble was adopted for the simulation system to ensure a constant total energy. Periodic boundary conditions were chosen as the boundary conditions to reduce the influence of boundary effects. The time step was set to 0.25 fs. To study the influence of the electric field on interfacial electrochemical action, a constant electric field intensity was applied to the model. By adjusting the magnitude of the electric field intensity, the influence laws of the electric field on the interfacial reactions of atoms on the SiC surface and the material removal process were investigated.

Due to minimal surface reaction changes observed within the 200 ps simulation timeframe under applied electric fields alone, comparative analyses of electric field intensity effects were conducted under elevated temperature conditions. Although SiC exhibits no reactivity with water molecules under ambient conditions, high-temperature environments (>2000 K) enable interfacial reactions by providing sufficient atomic kinetic energy to overcome the reaction barrier. Notably, the applied electric field further enhances reactivity by both lowering the effective activation energy and reducing the required reaction temperature.

## 3. Results

### 3.1. Corrosion Behavior of SiC Polytypes

Potentiodynamic polarization tests were conducted on four different types of SiC, and the polarization curves are shown in Figure 3a. It could be seen that there were obvious passivation regions in the electrochemical corrosion of all four types of SiC. The self-corrosion potentials obtained from the polarization curves are shown in Figure 3b. The self-corrosion potential of the 4H-4° sample was −0.953 V, while that of the 6H-4° sample was −0.972 V. The 4H-0° sample had a self-corrosion potential of −0.965 V, and the 6H-0° sample’s self-corrosion potential was −0.975 V. Evidently, the self-corrosion potentials of all samples showed minimal variation, indicating comparable corrosion characteristics among the four SiC types.

The surface roughness of four SiC samples after the potentiodynamic polarization test is shown in Figure 3c. The initial surface roughness averaged 1.5 nm across all samples. After corrosion, the surface roughness of the 4H-4° sample increased to 7.17 nm, while that of the 6H-4° sample increased to 2.26 nm. The 4H-0° sample exhibited a surface roughness of 8.2 nm, while the 6H-0° sample reached 3.33 nm. These results indicated that all SiC variants experienced surface degradation through corrosion and oxidation processes during the potentiodynamic polarization test. The 4H polytype exhibited approximately four times greater roughness increase than 6H samples, indicating that the crystallographic structure dominated over surface inclination in determining corrosion susceptibility. For subsequent mechanistic studies, the 4H-0° sample was selected due to its most pronounced surface modification.

### 3.2. Corrosion Behavior of SiC with Different Electrolyte Concentrations

The polarization curves of SiC substrates in NaCl solutions with different concentrations were obtained through potentiodynamic polarization tests, as shown in Figure 4a. All tested concentrations exhibited clear passivation regions during SiC corrosion, though with notable variations in self-corrosion potentials. As shown in Figure 4b, the measured self-corrosion potentials demonstrated concentration-dependent behavior: −0.931 V (0.2 mol/L), −0.941 V (0.4 mol/L), −0.932 V (0.6 mol/L), 0.684 V (0.8 mol/L), −0.656 V (1.0 mol/L), and 0.407 V (1.2 mol/L). Notably, the self-corrosion potentials remained relatively stable and significantly higher for electrolyte concentrations between 0.2–0.6 mol/L, suggesting accelerated corrosion rates within this concentration range.

The surface roughness of SiC samples corroded in electrolytes with different concentrations is shown in Figure 4c. Before corrosion, the initial surface roughness of the samples was approximately 1.5 nm. After corrosion, the surface roughness increased to 5.34 nm (0.2 mol/L), 5.63 nm (0.4 mol/L), 7.64 nm (0.6 mol/L), 3.81 nm (0.8 mol/L), 3.80 nm (1.0 mol/L), and 5.83 nm (1.2 mol/L) in respective electrolyte concentrations. Evidently, the maximum roughness change (7.64 nm) occurred at a 0.6 mol/L concentration, demonstrating the most aggressive corrosion rate, which was consistent with the results of the polarization curves.

The Raman spectroscopy results on the surface of the SiC sample before and after electrochemical corrosion in a 0.6 mol/L NaCl solution are shown in Figure 5. A Raman peak attributed to SiC was observed at 994 cm^−1^. After fitting and peak separation of the original curve, the positions of the new characteristic peaks of SiC and silicon oxide were at 985.6 cm^−1^ and 1022.1 cm^−1^, respectively. By comparing the Raman spectra before and after corrosion, it could be clearly seen that the color of the detection area changed significantly. Moreover, the characteristic peak of silicon oxide in the spectrum after corrosion showed notable enhancement in both intensity and peak width. This phenomenon provided clear evidence of silicon oxide formation through electrochemical corrosion of SiC in a 0.6 mol/L NaCl solution. Based on these findings and considering the relatively rapid corrosion rate observed at this concentration, a 0.6 mol/L NaCl solution was selected as the standard electrolyte for subsequent experiments investigating the electrochemical corrosion mechanisms of SiC substrates.

### 3.3. Corrosion Behavior of SiC at Different Voltages

Following the electrolyte selection, the optimal anodic polarization potential was investigated by applying potentiostatic polarization at voltages ranging from 2 V to 8 V. The resulting current density curves (Figure 6a) revealed a voltage-dependent increase in corrosion activity. The measured current densities were 0.15, 2.16, 4.09, and 8.69 mA/cm^2^ at applied voltages of 2, 4, 6, and 8 V, respectively. It could be found that the current density exhibited a stable and consistent upward trend with increasing voltage throughout the polarization period, indicating active dissolution behavior without passivation within the 2–8 V range.

The surface morphologies and surface roughness of SiC samples at different corrosion voltages are presented in Figure 6b. The surface roughness of the samples before corrosion was approximately 1.5 nm. Following electrochemical corrosion, minimal roughness variation was observed at both 2 V and 6 V, while moderate increases to 2.11 nm and 5.72 nm were recorded at 4 V and 8 V, respectively. Notably, the roughness fluctuations between 2–6 V remained within the range of inherent surface variations, suggesting these voltages induced negligible morphological modification. A significant roughness transition occurred only at 8 V, where an approximately three times increase relative to the initial state confirmed substantial corrosion-induced surface alteration.

The corrosion behavior of SiC substrates at elevated voltages (>10 V) was characterized using a DC power supply, with the voltage-dependent current density profiles presented in Figure 7a. The data revealed a monotonic increase in current density with applied voltage up to 25 V, followed by a marked reduction in growth rate within the 25–30 V range. This transition suggested the onset of passivation layer formation, which likely inhibited the electrochemical corrosion process at higher potentials.

Figure 7b displayed the surface morphology evolution and corresponding roughness measurements of SiC samples subjected to electrochemical corrosion at various voltages. The pristine samples exhibited an initial surface roughness of 1.5 nm. The surface roughness demonstrated a pronounced voltage dependence, escalating from 2.93 nm (10 V) to 26.15 nm (15 V) and peaking at 179.85 nm (20 V), followed by a subsequent reduction to 150.2 nm (25 V) and a dramatic decline to 23.64 nm (30 V). This characteristic of voltage-dependent roughness revealed three distinct corrosion regimes. The initial linear increase from 10–20 V corresponded to the active dissolution region where corrosion progressed unimpeded. The subsequent reduction between 20 and 25 V marked the transition into the stable passivation region, where protective surface layers began forming. Finally, the dramatic decrease in roughness from 25–30 V clearly demonstrated that passive layer dissolution became the dominant process at higher potentials.

### 3.4. Reactive Molecular Dynamics Simulation of Corrosion Behavior on SiC Surface

Reactive molecular dynamics simulations were performed to investigate the interfacial chemical reactions between 4H-SiC and H_2_O molecules under combined thermal (293–3293 K) and electric field (0–0.05 V/Å) conditions, with periodic boundary conditions applied in all directions. The electric field was applied along the z direction from bottom to top, establishing the SiC as the anode. The system temperature was linearly increased from 293 K to 3293 K within 200 ps. The simulation processes under different electric field intensities are shown in Figure 8. The heating process revealed distinct temperature-dependent evolution in both the atomic structure of SiC and its interfacial reactions with water molecules. When the temperature rose to 1643 K, localized deformation initiated in the surface atomic structure, accompanied by partial transformation of SiC into an amorphous phase. Further heating to 2543 K triggered the penetration of oxygen and hydrogen species into the SiC lattice. Upon reaching 2993 K, the SiC lattice was basically amorphized except for the fixed layer and the constant-temperature layer, and a large number of oxygen and hydrogen atoms penetrated into the SiC lattice. These structural transformations demonstrated progressively intensified interfacial reactions between SiC and water molecules with increasing temperature and simulation duration.

The temporal evolution of SiC amorphization under varying electric fields exhibited distinct field-dependent characteristics. As shown in Figure 9a, the number of amorphous atoms increased linearly with electric field intensity from 0 to 5 E^−4^ V/Å, reaching maximum amorphization, but decreased significantly when the field reached 0.05 V/Å. Figure 9b shows the penetration depth of external atoms into SiC at different electric field intensities. The data indicated that during the period of 0–90 ps, almost no atoms penetrated into SiC, and only partial amorphization occurred on the surface layer of SiC. During the period of 120–180 ps, the degree of surface amorphization increased continuously, which facilitated reactions with water molecules. Upon increasing the electric field intensity from 0 to 5 E^−4^ V/Å at 180 ps, both the penetration depth of external atoms into SiC and the corresponding surface corrosion depth exhibited significant enhancement. When the electric field intensity was 0.05 V/Å, the penetration depth of external atoms into SiC decreased.

## 4. Discussion

The experimental results of the corrosion characteristics of different types of SiC indicated that the lattice type played a dominant role in the corrosion process, while the inclination angle exhibited minimal influence. Specifically, the 4H-0° SiC variant demonstrated the highest corrosion rate. Furthermore, the electrolyte concentration significantly affected the corrosion behavior of SiC, with experimental data indicating peak corrosion rates occurring in 0.6 mol/L NaCl solution. Raman spectroscopy analysis confirmed the formation of silicon oxide on the SiC surface after corrosion.

The investigation of SiC corrosion behavior under varying voltages established critical parameters for optimizing electrochemical mechanical polishing processes. Experimental results showed that three distinct voltage-dependent regions: (1) an active dissolution region (2–20 V) where corrosion rates increased linearly with applied voltage, (2) a stable passivation region (20–25 V), and (3) a transition region (25–30 V) characterized by partial dissolution of surface oxide layers. Complementary reactive molecular dynamics simulations revealed that electric field strength significantly influenced atomic structure evolution during thermal processing. Within a certain range, an increase in the electric field strength would promote amorphization and reaction depth of atoms. However, when it exceeded a specific threshold, this process would be inhibited. This result was generally consistent with the results of electrochemical corrosion experiments, indicating that when the applied electric field strength exceeded a certain range, a passivation layer would form on the SiC surface, thereby hindering the continuous progress of surface reactions.

The electrochemical corrosion mechanism was illustrated in Figure 10a. Functional groups with high activity (OH^−^, O^2−^, and •OH), generated by the electrolysis of water during electrochemical corrosion, participated in the formation of a complex SiO_2_ modified layer on the SiC surface [41]. Figure 10b–d provides mechanistic insights into the electrochemical corrosion process. Under low-voltage conditions, surface defects on SiC, such as scratches and cracks, preferentially underwent oxidation reactions due to the instability of the atomic structure, leading to the aggravation of these defects. As the applied voltage increased and the electric field strength was enhanced, the oxidation reaction occurred more uniformly across the entire surface. At excessive voltages, the rapid formation of a dense passivation layer inhibited further oxidation, while extreme voltages induced passivation layer breakdown and subsequent oxide dissolution. This mechanistic elucidation is expected to establish a solid theoretical foundation for understanding the microscopic evolution of SiC during electrochemical corrosion, providing critical guidance for process optimization.

## 5. Conclusions

This study systematically investigated the electrochemical corrosion mechanism of SiC substrates through integrated electrochemical experiments and reactive molecular dynamics simulations. The corrosion behavior of the SiC substrate at the nanoscale was analyzed, and the corrosion reaction mechanisms were summarized. The following conclusions can be drawn:(1)The crystalline structure exhibited a dominant influence on the corrosion characteristics of the SiC substrate, while the inclination angle played a minimal role. The 4H-0° SiC substrate demonstrated the highest corrosion rate among the tested samples. Moreover, the peak corrosion rate was achieved using a 0.6 mol/L NaCl electrolyte solution.(2)The electrochemical corrosion behaviors of SiC substrates revealed three voltage-dependent characteristics: active dissolution (2–20 V), stable passivation (20–25 V), and transpassive dissolution (25–30 V). Reactive molecular dynamics simulations further revealed that both amorphization degree and penetration depth of external atoms on the SiC surface showed a decreasing trend at elevated voltages, suggesting a corresponding reduction in the corrosion rate when the voltage exceeded the optimal range.(3)OH^−^, O^2−^, and •OH generated by the electrolysis of water during electrochemical corrosion immediately reacted with the SiC surface to form a SiO_2_ modified layer. At lower voltages, preferential oxidation occurred at surface defects, accentuating existing scratches and cracks. Moderate voltages promoted uniform surface oxidation across the entire substrate. Elevated voltages induced passivation layer formation, effectively inhibiting further oxidation. Under extreme voltage conditions, the passivation layer underwent breakdown, leading to dissolution of the oxide layer. This comprehensive analysis provided fundamental insights into SiC electrochemical corrosion mechanisms, offering valuable guidance for material processing and applications.

## Figures and Tables

**Figure 1 micromachines-16-01129-f001:**
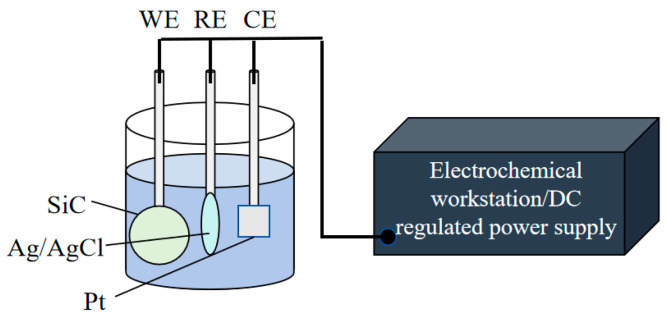
Schematic diagram of electrochemical workstation equipment.

**Figure 2 micromachines-16-01129-f002:**
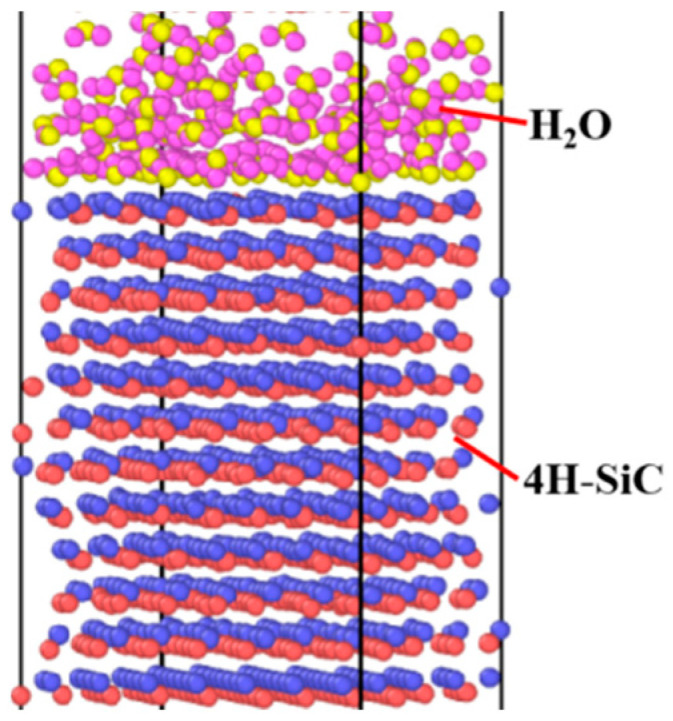
The model of reactive molecular dynamics simulation for SiC.

**Figure 3 micromachines-16-01129-f003:**
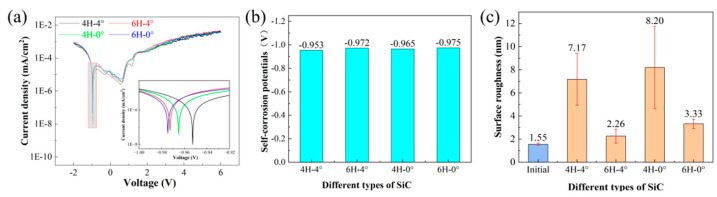
Potentiodynamic polarization test results of SiC polytypes: (**a**) potentiodynamic polarization curves; (**b**) self-corrosion potential; (**c**) surface roughness of SiC.

**Figure 4 micromachines-16-01129-f004:**
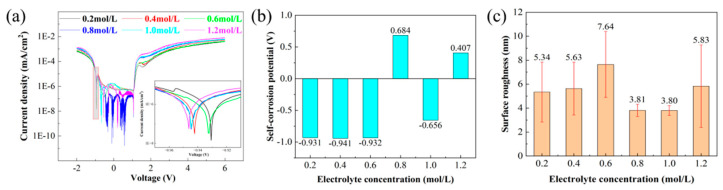
Potentiodynamic polarization test results of SiC with different electrolyte concentrations: (**a**) potentiodynamic polarization curves; (**b**) self-corrosion potential; (**c**) surface roughness of SiC.

**Figure 5 micromachines-16-01129-f005:**
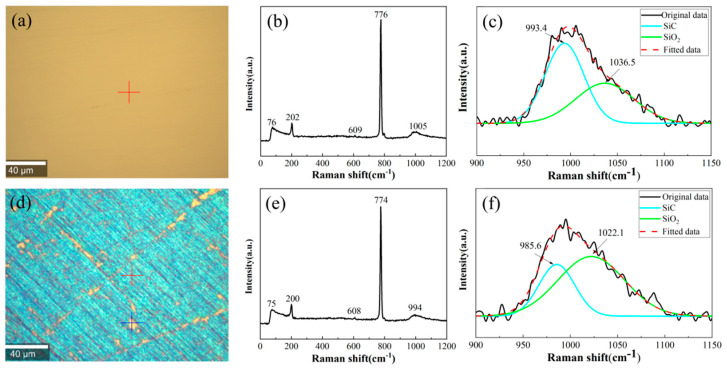
Raman spectral detection results on SiC surface: (**a**–**c**) before corrosion; (**d**–**f**) after corrosion in 0.6 mol/L electrolyte.

**Figure 6 micromachines-16-01129-f006:**
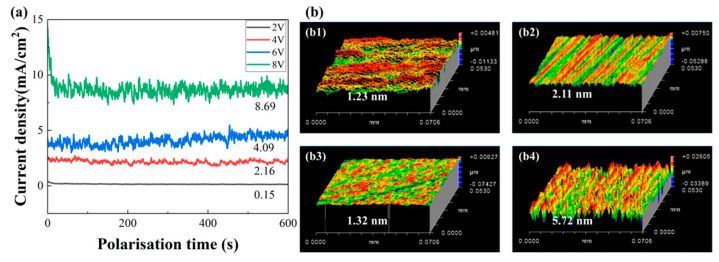
Constant potential polarization test results of SiC substrates at different corrosion voltages: (**a**) constant potential polarization curves; (**b**) surface morphologies and surface roughness of SiC: (**b1**) 2 V; (**b2**) 4 V; (**b3**) 6 V; (**b4**) 8 V.

**Figure 7 micromachines-16-01129-f007:**
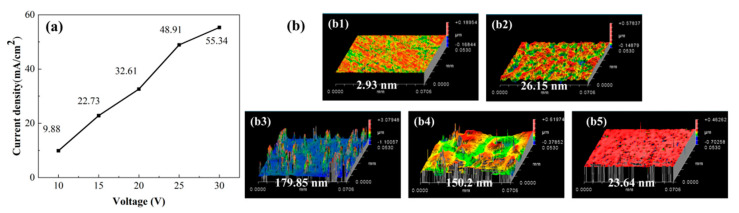
Constant potential corrosion results of SiC substrates at elevated voltages: (**a**) constant potential polarization curves; (**b**) surface morphologies and surface roughness of SiC: (**b1**) 10 V; (**b2**) 15 V; (**b3**) 20 V; (**b4**) 25 V; (**b5**) 30 V.

**Figure 8 micromachines-16-01129-f008:**
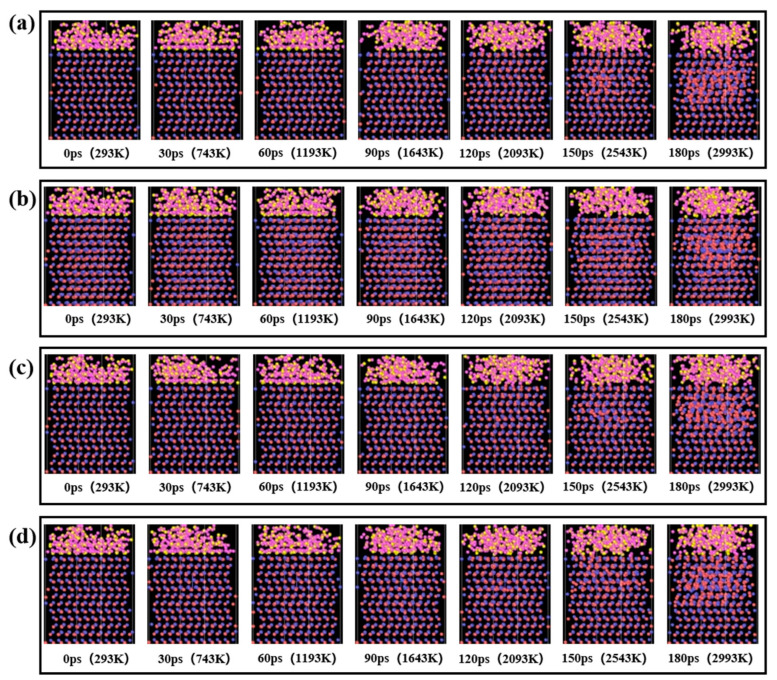
Simulation process under different electric field strengths: (**a**) 0 V/Å; (**b**) 5.0 E^−6^ V/Å; (**c**) 5.0 E^−4^ V/Å; (**d**) 5.0 E^−2^ V/Å.

**Figure 9 micromachines-16-01129-f009:**
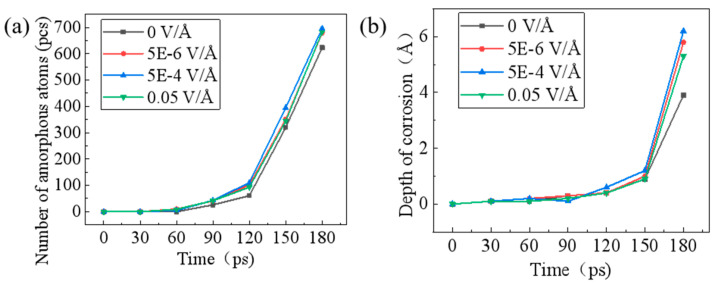
Number of atoms in amorphous state (**a**) and depth of corrosion (**b**) versus time at different electric field strengths.

**Figure 10 micromachines-16-01129-f010:**
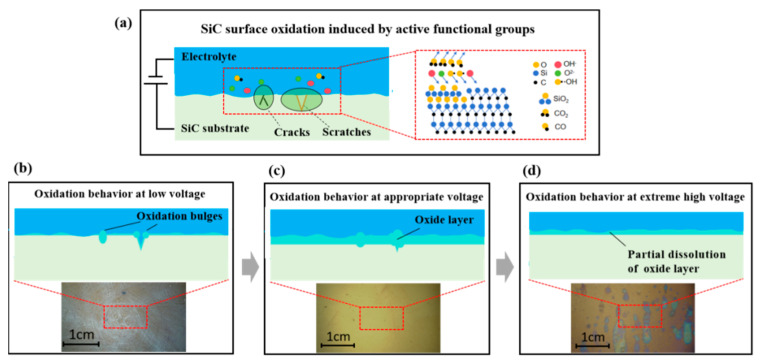
Electrochemical corrosion mechanism (**a**) and reaction processes (**b**–**d**) of SiC substrates at different voltages.

## Data Availability

The data presented in this study are available on request from the corresponding author.

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
