# Peer review of "Atomic-Scale Revelation of Voltage-Modulated Electrochemical Corrosion Mechanism in 4H-SiC Substrate"

_micromachines, 2025, doi:10.3390/mi16101129_

Round 1
Reviewer 1 Report
Comments and Suggestions for Authors
This manuscript deals with the dissolution behaviour of SiC polymorphs in NaCl under different conditions.
The elucidation of the dissolution mechanisms of SiC semiconductor materials is of high importance for production technologies. The autors claim in the abstract to contribute to this topic with comprehensive experiments and molecular dynamics simulations. However, all together, this work contributes only little to the open questions. In particular, it is unclear to what extent the simulation part can meaningfully complement the experiments, as the simulation was conducted for completely different conditions (temperatures as high as 3000K). Furthermore, the conclusions are not strongly supported by the experiments.
Detailed review:
Abstract: In the abstract it should be mentioned, that the SiC has the positive polarity of the voltage, especially because few lines later the authors refer to reaction products from the negative terminal ("Active functional groups (OH-, O2- and •OH), generated by the electrolysis of water...."). This is also mentioned several times in the rest of the text, which is not applicable for an anodic oxidation reaction).
lines 83pp: Authors cite a reference for water corrosion of SiC at above 1000K. This is not relevant here, since it is obviously corrosion in the gas phase.
line 103: "(1) a platinum sheet electrode clip served as the working electrode to secure the SiC sample, ..." the platinum sheet electrode clip is only the contact, the working electrode is the SiC.
line 107: make clear that the O-terminated surface was investigated.
line 121: the measured range was 0-1200 cm-1.
line 134: " ... is shown ..." -> the same in lines 157, 159, 167, 190, 201, 267
line 169: " 3.33 nm.." -> "3.33 nm." Two dots or double punctation marks appear also in lines 269 and 272.
paragraph lines 156pp: Probably a better parmeter for the influence of surface structure and concentration is the onset potential of the current at around +1V. It is usually related to pitting corrosion.
The 3D diagrams in Figs. 3 and 4 are of poor quality. They should be plotted in a regular 2D plot. In addition they are mixed up. In Fig.3 the current has a negative sign. The current scale in Fig.4 is completely different. The x-scales should be plotted in the same style.
The concentration-dependent behavior seems to be random (corrosion potential and roughness). It is commendable that the authors provided statistical data with mean values and standard deviations, however there is only low statistical evidence for that what they are claiming, for both, the polarization data and the roughness data.
line 200: "The Raman spectral detection results ... " -> "The Raman spectroscopy results ... "
In the discussion line 327 authors say: "This mechanistic elucidation established a solid theoretical foundation ... " I believe this statement is somewhat presumptuous.
line 340: " ... while the inclination angle played minimal influence." either " ... while the inclination angle played minimal role." or " ... while the inclination angle had minimal influence."
Author Response
Comments 1
- In particular, it is unclear to what extent the simulation part can meaningfully complement the experiments, as the simulation was conducted for completely different conditions (temperatures as high as 3000K). Furthermore, the conclusions are not strongly supported by the experiments.
Response: From the perspective of observation dimensions, experimental measurements focus on the overall average temperature of a macroscopic system after it reaches thermal equilibrium. In contrast, molecular dynamics simulations can measure local instantaneous temperatures, with a time scale only ranging from picoseconds to nanoseconds, which is insufficient to simulate the macroscopic thermal equilibrium temperature. Therefore, it is reasonable for a deviation to exist between the simulated instantaneous high temperature and the experimentally measured macroscopic temperature. Additionally, in terms of results, the compositions of surface reaction products of the SiC wafers was consistent between the experiment and simulation, with both containing SiO₂. This consistency serves to verify that the experiment and simulation can mutually corroborate each other.
- In the abstract it should be mentioned, that the SiC has the positive polarity of the voltage, especially because few lines later the authors refer to reaction products from the negative terminal ("Active functional groups (OH-, O2- and •OH), generated by the electrolysis of water...."). This is also mentioned several times in the rest of the text, which is not applicable for an anodic oxidation reaction).
Response: The electrochemical oxidation of SiC relies on electron and ion migration. By applying an external voltage to the SiC surface, anions generated from water electrolysis migrate toward the anode, which in turn induces anodic oxidation and ultimately leads to the formation of an oxide layer. Although SiC exhibits high chemical inertness, it can still be oxidized under an appropriate electric field[1]. In addition, We have highlighted the textual expressions regarding silicon carbide (SiC) as an anode, which are marked in red in the abstract.
References: Wu, P.; Zhao, D.; Liu, N.; et al. Effect of voltage on electrochemical mechanical polishing (ECMP) of 4H–SiC with fixed abrasives, Journal of Materials Research and Technology, 2025, 36, 7807-7817.
- lines 83pp: Authors cite a reference for water corrosion of SiC at above 1000K. This is not relevant here, since it is obviously corrosion in the gas phase.
Response: In non-reactive molecular dynamics (MD), regardless of water's macroscopic state (liquid, solid, or gas), it primarily exists and functions in the form of H2O molecules, with no chemical bond breaking or formation involved. In contrast, reactive MD enables water to dissociate into ions (e.g., H+, OH-) or free radicals (e.g., H-, O-), and such dissociation occurs under extreme macroscopic conditions (e.g., high temperature, high pressure) or during chemical reactions. Therefore, the cited reference is reasonable.
- Theplatinum sheet electrode clip is only the contact, the working electrode is the SiC.
Response: According to the Reviewer’s comments, we have revised as “A SiC sample secured by a platinum sheet electrode clip served as the working electrode” in Experimental details which is marked in red in this manuscript.
- line 107: make clear that the O-terminated surface was investigated.
Response: According to the Reviewer’s comments, we have added “The SiC samples have an initial surface roughness of approximately 1.5 nm (Ra), with no obvious oxidation on the surface.” in Experimental details which is marked in red in this manuscript.
- The measured range was 0-1200 cm-1.
Response: We have changed it in Experimental details which is marked in red in this manuscript.
- " ... is shown ..." -> the same in lines 157, 159, 167, 190, 201, 267
Response:We have revised it in this manuscript.
- " 3.33 nm.." -> "3.33 nm." Two dots or double punctation marks appear also in lines 269 and 272.
Response:We have revised it in this manuscript.
- 9. paragraph lines 156pp: Probably a better parmeter for the influence of surface structure and concentration is the onset potential of the current at around +1V. It is usually related to pitting corrosion.
Response: Section 3.1 focuses on the selection of silicon carbide (SiC) type employed in this study. The self-corrosion potentials reflects the "driving force" for the spontaneous corrosion of a material in a given environment. Generally, the more negative the self-corrosion potential, the higher the tendency of the material to undergo corrosion. As shown in Figures 3 a and b, the self-corrosion potentials of the four SiC samples were approximately identical. Subsequently, potentiodynamic polarization tests were conducted on these four four types of SiC samples, as shown in Figure 3 c. The results indicated that the 4H-0° sample exhibited the highest corrosion susceptibility, with the most significant variation in surface roughness after corrosion. To investigate the electrochemical corrosion mechanism of SiC more clearly, the 4H-0° SiC sample was selected for subsequent experiments.
- 10. The 3D diagrams in Figs. 3 and 4 are of poor quality. They should be plotted in a regular 2D plot. In addition they are mixed up. In Fig.3 the current has a negative sign. The current scale in Fig.4 is completely different. The x-scales should be plotted in the same style.
Response: As shown in the figure below, we previously attempted to plot the experimental results in the form of a 2D graph. However, due to the overlap of the curves in the generated graph, we chose to use a 3D graph instead, as it offers better readability. In addition, we have revised the issues concerning the coordinate scales in Figure 3 and Figure 4.
- 11. The concentration-dependent behavior seems to be random (corrosion potential and roughness). It is commendable that the authors provided statistical data with mean values and standard deviations, however there is only low statistical evidence for that what they are claiming, for both, the polarization data and the roughness data.
Response: In fact, the experimental results exhibit regularity. As shown in Figure 4, within the low concentration range of 0.2~0.6 mol/L, the change in the sample's surface roughness is comparable to, or even more pronounced than, that within the high concentration range of 0.8~1.2 mol/L. This indicates a relatively intense corrosion rate at low concentrations, which is consistent with the results of the polarization curves. At the high concentration of 1.2 mol/L, although the self-corrosion potential is relatively high, the significant change in surface roughness is attributed to the high concentration of the electrolyte.
- "The Raman spectral detection results ... " -> "The Raman spectroscopy results ... "
Response: We have revised it in this manuscript.
- In the discussion line 327 authors say: "This mechanistic elucidation established a solid theoretical foundation ... " I believe this statement is somewhat presumptuous.
Response: According to the Reviewer’s comments, we have revised as “This mechanistic elucidation is expected to establish a solid theoretical foundation for understanding the microscopic evolution of SiC during electrochemical corrosion, providing critical guidance for process optimization.” in Discussion which is marked in red in this manuscript.
- " ... while the inclination angle played minimal influence." either " ... while the inclination angle played minimal role." or" ... while the inclination angle had minimal influence."
Response: According to the Reviewer’s comments, we have revised as “...,while the inclination angle played minimal role.” in Conclusions which is marked in red in this manuscript.

Reviewer 2 Report
Comments and Suggestions for Authors
It is well known that SiC substrate surface quality is important for the device performance, operational reliability, and service lifetime. In this work the comprehensive experiments and molecular dynamics simulations were used for observing the electrochemical corrosion behavior of SiC substrate. It was found that under the current researches on SiC ECMP technology, the general electrochemical corrosion mechanism of SiC substrate was to be recognized and understood.
This work – after Introduction – covered experimental part, the study results, a part devoted for discussion, and three critical conclusions. The paper covers 10 Figures, lacking Figure 1 in the text (no Tables provided). In References (42 in total), all of them were useful, with [42] …RSC Advances.
The corrosion behavior of SiC substrate in nanoscale was analyzed. SiC electrochemical corrosion mechanisms were studied, indicating the way for material processing and applications.
To sum up, after a minor correction is done – this work could be considered for publication.
Author Response
Comments 2
- The paper covers 10 Figures, lackingFigure 1 in the text (no Tables provided). In References (42 in total), all of them were useful, with [42] …RSC A
Response: According to the Reviewer’s comments, we have added “The experimental setup of the electrochemical workstation is shown in Figure 1.” in Experimental details which is marked in red in this manuscript. In addition, we have revised the issues concerning citation format of reference 42.

Round 2
Reviewer 1 Report
Comments and Suggestions for Authors
The authors improved their manuscript and corrected several relevant errors.
However, some shortcomings of the manuscript mentioned in my first comments have not been adequately adressed.
1.) Simulation part: Authors argue that this kind of MD simulation is relevant for the interpretation of their experimental results. I still do not believe this, but I think that the judgement about the significance of the simulation calculations can be left to the reader.
2) Products from the negative terminal ("Active functional groups (OH-, O2- and •OH). In the reference given by the authors (Wu, P. et.al.), the cathodic product ·OH is generated and active at the anode under very specific circumstances: H2O2 containing polishing slurry (which generates the OH radicals cathodically) and very close distance between cathode and anode. All these conditions are not given in the experiments described in this manuscript and therefore I dont belive that ·OH (and O2-) play a significant role. Only after very long polarisation and larger currents an alkalization of the solution (increase of OH-) might be observed. Since the statement(s) about "active functional groups" is misleading, please omit.
-> By the way: In the experimental section, the information about the scan rate for the potentiodynamic polarization tests is missing. Please add!
3.) Again, I doubt the significance of water corrosion of SiC at above 1000K for corrosion at room temperature in NaCl solution, especially since the corrosion mechanisms changes at high temperatures and with oxidizing conditions, but as mentioned before, the judgement can be left to the reader.
5.) "make clear that the O-terminated surface was investigated." This was a mistake on my part, sorry for that! I wanted to know if the C-terminated or the Si-terminated face was investigated. Therefore the authors responded ineligibly. Please, indicate which side of the SiC was taken for the experiments because it might have an impact on the interpretation of the OCPs.
10.) regarding the 3D diagrams in Figs. 3 and 4. Authors show in their response the 2D graph as a bad example and argue: "due to the overlap of the curves in the generated graph, we chose to use a 3D graph instead". I see that completely different! The 2D graph gives much more information than the 3D one. I agree, the 3D plot looks nice, it may fit to a graphical abstract or a popular magazine but not to a scientific study. I strongly recommend to switch to the 2D diagram as shown in the author response!
After consideration of the comments above I can recommend the publication of this mansucript.
Author Response
- Simulation part: Authors argue that this kind of MD simulation is relevant for the interpretation of their experimental results. I still do not believe this, but I think that the judgement about the significance of the simulation calculations can be left to the reader.
Response: The results of molecular dynamics simulations associated with nanomachining are sometimes questioned, as it is difficult to compare and validate them through fully corresponding experiments. In this study, experiments and simulations were compared and validated in terms of products and the relationship between products and variables, where both exhibited similar trends and phenomena. Therefore, the consistency between the experimental and simulation results demonstrates that the molecular dynamics simulation model and method employed in this research are reliable.
- Products from the negative terminal ("Active functional groups (OH-, O2- and •OH). In the reference given by the authors (Wu, P. et.al.), the cathodic product •OH is generated and active at the anode under very specific circumstances: H2O2 containing polishing slurry (which generates the OH radicals cathodically) and very close distance between cathode and anode. All these conditions are not given in the experiments described in this manuscript and therefore I dont believe that •OH (and O2-) play a significant role. Only after very long polarisation and larger currents an alkalization of the solution (increase of OH-) might be observed. Since the statement(s) about "active functional groups" is misleading, please omit.By the way: In the experimental section, the information about the scan rate for the potentiodynamic polarization tests is missing. Please add!
Response: According to the Reviewer’s comments, we have removed the misleading term "active functional groups" in this manuscript. Becides, we have added “The potentiodynamic polarization measurements were conducted at a scan rate of 10 mV/s.” in Experimental details which is marked in red in this manuscript.
- Again, I doubt the significance of water corrosion of SiC at above 1000K for corrosion at room temperature in NaCl solution, especially since the corrosion mechanisms changes at high temperatures and with oxidizing conditions, but as mentioned before, the judgement can be left to the reader.
Response: Considering the actual environmental factors of the experiment, the simulated temperature is much higher than the experimental temperature. This phenomenon explains why SiC can undergo water corrosion at room temperature in experiments, while in simulations, water corrosion of SiC occurs at temperatures exceeding 1000 K. Furthermore, combined with the fact that the experiment and simulation yield the same products, there is a strong correlation between the experimental results and the simulation outcomes.
- Please, indicate which side of the SiC was taken for the experiments because it might have an impact on the interpretation of the OCPs.
Response: According to the Reviewer’s comments, we have revised as “Experiments were conducted on the Si-face (0001) of SiC samples, with an initial surface roughness of approximately 1.5 nm (Ra) and no obvious oxidation on the surface.” in Experimental details which is marked in red in this manuscript.
- I strongly recommend to switch to the 2D diagram as shown in the author response!
Response: According to the Reviewer’s comments, we have switched potentiodynamic polarization curves to a 2D graph in Figure 3 and Figure 4 of this manuscript.